# High Resting Heart Rates Are Associated with Early Posthospitalization Mortality in Low Ejection Fraction Patients

**DOI:** 10.3390/jcm11102901

**Published:** 2022-05-20

**Authors:** Andreas Hain, Nikolai Busch, Said Elias Waezsada, Julie Hutter, Patrick Kahle, Malte Kuniss, Thomas Neumann, Tsyuoshi Masuda, Horst O. Esser, Christian Hamm, Johannes Sperzel

**Affiliations:** 1Kerckhoff Klinik Bad Nauheim, 61231 Bad Nauheim, Germany; n.busch@kerckhoff-klinik.de (N.B.); s.waezsada@kerckhoff-klinik.de (S.E.W.); j.hutter@kerckhoff-klinik.de (J.H.); p.kahle@kerckhoff-klinik.de (P.K.); m.kuniss@kerckhoff-klinik.de (M.K.); t.neumann@kerckhoff-klinik.de (T.N.); j.sperzel@kerckhoff-klinik.de (J.S.); 2ZOLL Services LLC, Pittsburgh, PA 15238, USA; tmasuda@zoll.com (T.M.); hesser@zoll.com (H.O.E.); 3Department of Cardiology, University Hospital Giessen, 35392 Giessen, Germany; c.hamm@kerckhoff-klinik.de

**Keywords:** heart failure, WCD, heart rate control, GDMT, beta blocker, sudden cardiac death

## Abstract

Guideline-directed medical therapy (GDMT) is crucial in reducing mortality in patients with heart failure with heart rate lowering by a beta blocker (BB) being an important therapeutic concept. We aimed to assess the usefulness of a wearable cardioverter/defibrillator (WCD) to provide detailed information about heart rate for managing patients with reduced left ventricular ejection fraction (LVEF) and symptoms of heart failure and to correlate mortality with the mean heart rate. A total of 4509 consecutive patients (mean age: 59 + 13 years, 88% male) were analyzed retrospectively. All patients had reduced LVEF and were prescribed a WCD for protection from sudden cardiac death (SCD) during GDMT uptitration awaiting LVEF recovery. The device continuously measured nighttime and daytime HR at the beginning and end of WCD use. Patients who died during wear time had significantly higher HRs compared with survivors: daytime beginning of use (BOU), 80 ± 15 bpm vs. 76 ± 13, *p* < 0.01; nighttime BOU, 76 ± 14 vs. 69 ± 13, *p* < 0.0001; daytime end of use (EOU), 84 ± 20 vs. 73 ± 13, *p* < 0.0001; nighttime EOU, 80 ± 20 vs. 65 ± 12, *p* < 0.0001). In conclusion, HR monitoring with a WCD yields important prognostic information and may assist in optimal usage of BB in patients with low LVEF.

## 1. Introduction

Guideline-directed medical therapy (GDMT) represents the cornerstone of management of heart failure (HF) patients. The essential components of this therapy as described in international guidelines are ACE inhibitors (ACE-I), ARN inhibitors (ARNI), RASS inhibitors (RAAS-I), mineralocorticoid receptor agonists (MRAs), sodium–glucose cotransporter-2 (SGLT-2), and beta blockers (BBs), which are able to reduce rehospitalization and/or mortality [1,2]. Current guidelines suggest that a heart rate (HR) of <70 bpm is associated with better outcome in CHF patients [1]. Elevated HR is associated with cardiovascular mortality in the general population, in patients with ischemic heart disease [3], and in CHF patients.

Earlier studies have shown that therapy in patients suffering from congestive heart failure (CHF) with BB alone is able to reduce morbidity and mortality to a greater degree than single RAAS therapy [4,5]. Furthermore, BB offers an improvement of systolic left ventricular ejection fraction (sLVEF) and a reduction in the risk for sudden cardiac death (SCD) [6,7,8]. Similar data are available also for ACE/ARB and aldosterone antagonists. As the direct effect of single components of GDMT can only be measured for HR-controlling medication, heart rate provides a parameter that can be easily observed and guide optimization of medication.

Newly diagnosed HF with reduced ejection fraction is associated with an elevated risk for SCD [9]. Studies have shown that this risk is strongly linked to the reduced EF and may be only temporary as the EF can recover through GDMT [10].

The wearable cardioverter defibrillator (WCD) offers temporary protection against ventricular arrhythmias [10,11]. It also has several diagnostic features that enable the treating health-care professional to continuously monitor several vital parameters, such as HR, activity, and body position.

The purpose of this retrospective study was to correlate outcomes in the early posthospitalization period in ambulatory patients with low EF with WCD-derived HR measurements using the Zoll LifeVest (Pittsburgh, PA, USA) database.

## 2. Materials and Methods

### 2.1. Patients

Patients with clinical presentation of heart failure due to ischemic or nonischemic cardiomyopathy with a reduced LVEF of <35% were prescribed a WCD at the time of hospital discharge. Consecutive patients who were prescribed a WCD between 2015 and 2017 were included in this retrospective analysis. To be included in the dataset, patients had to have a minimum HR recording time of >250 min between 7:00 a.m. and midnight (daytime) or >100 min between midnight and 7:00 a.m. (nighttime) for at least 1 day during the first 3 calendar days of WCD wear and for at least 1 day during the last 3 calendar days of WCD wear. These time zones were chosen because of the patient’s reduced activity between midnight and 7:00 a.m. based on measured activity levels (step count) by the device. The requirements of >250 and >100 min minimum HR recording times were necessary to provide sufficient data points for the analysis. In addition, outcomes of patients regarding all-cause mortality, ICD implantation, and EF improvement had to be available. All patients consented to the data collection.

### 2.2. Device Description

The WCD (LifeVest^®^, Zoll, Pittsburgh, PA, USA) is composed of four dry nonadhesive capacitive electrodes and three dry-to-wet nonadhesive defibrillation electrodes incorporated into a chest strap assembly, along with a 1.7 lb defibrillator unit carried on a waist belt. The monitoring electrodes are positioned circumferentially around the chest and held in place by about 1 to 1.5 lb of tension. The defibrillation electrodes are positioned for apex-posterior defibrillation *(*Figure 1).

If an arrhythmia is detected, an escalating alarm sequence starts, including a vibration against the skin, audible tones, and a voice cautioning bystanders of an impending shock. Patients are trained to hold a pair of response buttons during these alarms. Responding acts as a test of consciousness: if no response occurs, the device extrudes gel from the defibrillation electrodes and delivers up to five 150-joule biphasic shocks. A detailed description of the WCD is described in previous publications and reviews [10,11].

The WCD also provides continuous monitoring of the ECG and heart rate. Whenever the patient wears the device, the HR is continuously measured, and the value for 5 min intervals is provided and stored. The heart rate derives from the measured ECG.

### 2.3. Data Analysis

Heart rate was measured continuously by the WCD. For the purpose of this analysis, we used the heart rate measured at the beginning and the end of WCD use; the change between HR at the beginning of use and end of use is described as HR trend. Beginning of WCD use (BOU) was defined as the first day during the first 3 days of WCD use where the device was worn for >250 min for daytime or >100 min for nighttime. End of WCD use (EOU) was defined as the last day during the last 3 days of WCD use where the device was worn for >250 min for daytime or >100 min for nighttime. Daytime and nighttime median HRs were calculated for each patient at BOU and EOU. Patients were grouped by survival status (alive or dead) at EOU. Patients who received an appropriate shock (shock delivered for ventricular tachycardia or ventricular fibrillation) were counted as alive. Descriptive statistics were used to summarize demographic clinical characteristics and diagnosis data. Categorical variables were expressed as numbers and percentages. All continuous variables are presented as mean ± SD unless indicated otherwise. A *t*-test was used to determine the significance in HR between the death and alive groups at BOU and EOU. *p*-Values < 0.05 were considered statistically significant. All analyses were conducted using R (version 3.6.1).

## 3. Results

A total of 4509 patients were included for analysis of daytime HR and nighttime HR at BOU and EOU (Table 1). The mean age was 59 ± 13 years, and 80% of the patients were male. A total of 88 patients (2%) died during the study period. The median length of WCD use was 66 days (IQR: 43–90).

Patients who died had significantly higher HRs compared with survivors (daytime BOU, 80 ± 15 bpm vs. 76 ± 13, *p* < 0.01; nighttime BOU, 76 ± 14 vs. 69 ± 13, *p* < 0.0001; daytime EOU, 84 ± 20 vs. 73 ± 13, *p* < 0.0001; nighttime EOU, 80 ± 20 vs. 65 ± 12, *p* < 0.0001) (Figure 2 and Table 2).

While the daytime and nighttime mean HRs dropped significantly (*p* < 0.0001) from beginning to end of WCD use in the group of patients surviving to the EOU, the daytime and nighttime HRs increased in the group of patients who died during wear time.

Of the patients who died, 22% and 69% had nighttime HR > 90 bpm or > 70 bpm at EOU, respectively (Table 3). In contrast to the patients who died, fewer survivors had elevated nighttime HRs at EOU (3% of the patients with HR > 90 bpm and 29% of the patients with HR > 70 bpm).

Among the 4509 patients, 4078 had sufficient nighttime HR data at both BOU and EOU to compare HR changes throughout the observed time frame. The HR trend over WCD wearing period was also associated with mortality (Figure 2 and Table 4).

Patients who had persistently higher HR (HR > 70 bpm at BOU and EOU, high–high) and whose HR increased over WCD wearing period (HR < 70 bpm at BOU, then HR > 70 bpm at BOU, low–high) had higher death rates of 3.8% and 4.9%, respectively (all *p* < 0.0001). In contrast, patients who had persistently lower HR (HR < 70 bpm at BOU and EOU, low–low) and whose HR decreased over time (HR > 70 bpm at BOU, then HR < 70 bpm at BOU, high–low) had lower death rates of 0.7% and 1.2%, respectively (all *p* < 0.0001). By applying the conventional definition of tachycardia (>90 bpm) [6], patients who had trends of high–high and low–high had higher death rates of 8.9% and 10.3%, respectively, compared with the trends with a threshold of 70 bpm (Figure 3 and Table 4).

## 4. Discussion

This retrospective analysis based on HR assessment in patients fitted with a WCD yields several important observations. In line with prior studies, the results demonstrate that an elevated HR (>70 bpm) is associated with increased all-cause mortality rate in HF patients with an LVEF of <35%. This was demonstrated in the present population of WCD patients for the early phase after hospitalization (BOU) and for the follow-up period of 3 months (EOU). More specifically, there was a higher all-cause mortality for those subjects whose average resting HR increased during the WCD monitoring period. Conversely, patients whose average resting HR decreased during the monitoring period had a more favorable outcome particularly when the average HR was <70 bpm. The present study, therefore, indicates that WCD-derived measurements of HR represent a useful tool to optimize therapy in HF patients with reduced LVEF.

Since the present data analysis is based on retrospective data, the data derived from the WCD could not be used to optimize therapy, implying that our patient cohort is a native cohort.

The benefits of heart rate control have been described in previous trials [6,7]. However, it is also known that heart rate control—even though it is a class I recommendation in the actual guidelines—is only achieved in a portion of all patients requiring this therapy. While immediately after the approval of BB for patients with HF in 1997 the acceptance was quite low, it has been shown in recent publications that still today optimal heart rate is only achieved in 57% of all patients requiring HR control [12].

The reasons for not achieving the optimal dosage of BB could be age, lack of patient compliance, comorbidities, and unwanted side effects (hypotension). In a recently published meta-analysis, no relationship between BB dosage and clinical outcome could be found [5]. The objective of HF treatment with BB cannot be measured by the optimal dosage of BB but needs to take into consideration that the desired treatment effect (HR < 70 bpm) needs to be controlled and the dosage adapted accordingly. If optimal HR cannot be achieved by BB alone, the current guidelines recommend ivabradine (in patients with SR). However, the underlying desired treatment effect needs to be taken into consideration. As it is not feasible to constantly monitor HR in outpatients, the optimal dosage of HR control medication remains a challenge.

While other HF medication is recommended for a maximum tolerated daily target dose, treatment with BB should follow the desired effect of heart rate control. Therefore it is necessary to have reliable information about daily heart rate and resting heart rate. One feature of the WCD is constant monitoring of HR and, therefore, enabling the treating physician to individualize patient therapy.

The advantage of the WCD is assessing prognostic implications of HR control (i.e., simultaneous protection from SCD, assessment of further WCD-derived prognostic indices, etc.). The question remains to what extent the measured data of the WCD can be used to achieve adequate HR control. One possible solution could be integration into existing telemonitoring care concepts and inclusion in predetermined treatment algorithms. Previous studies have shown that telemonitoring concepts do achieve survival benefit if—in addition to the pure collection of data—clear operating procedures, including timing of guidelines, have been installed based on the measured parameter [13].

The implementation or establishment of such patient management approaches can be time-consuming, so it makes sense to find solutions that can be incorporated into clinical practice. While the HR differences reported here are group results, focusing attention on patients not achieving an important physiologic effect and remaining as HR outliers may provide outcome benefit.

## 5. Conclusions

Our study in a contemporary sample of HF patients confirms prior observations on prognostic implications of HR. The WCD provides an excellent tool to assess HR changes over time in order to risk-stratify patients with reduced systolic function. At the same time, the WCD will protect these high-risk patients from serious ventricular tachyarrhythmias and sudden death. Future studies need to evaluate whether heart rate optimization, as assessed by the WCD, will improve prognosis in this patient population.

## Figures and Tables

**Figure 1 jcm-11-02901-f001:**
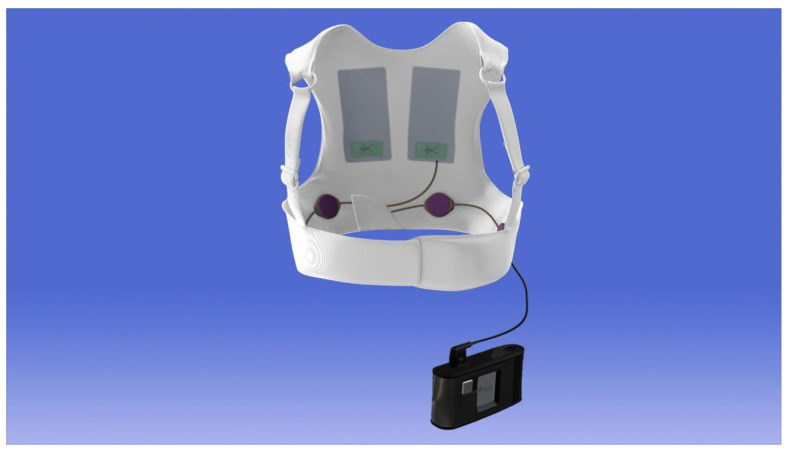
Wearable cardioverter defibrillator, picture courtesy of Zolll Services LLC, Pittsbugh, PA, USA.

**Figure 2 jcm-11-02901-f002:**
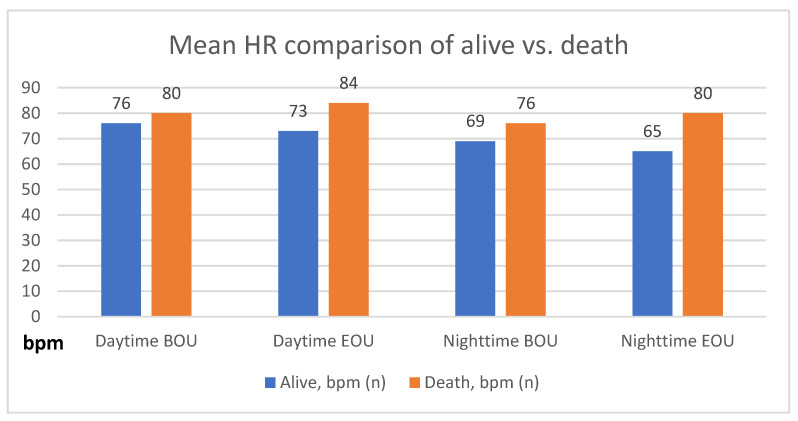
Mean HR comparison of alive vs. dead.

**Figure 3 jcm-11-02901-f003:**
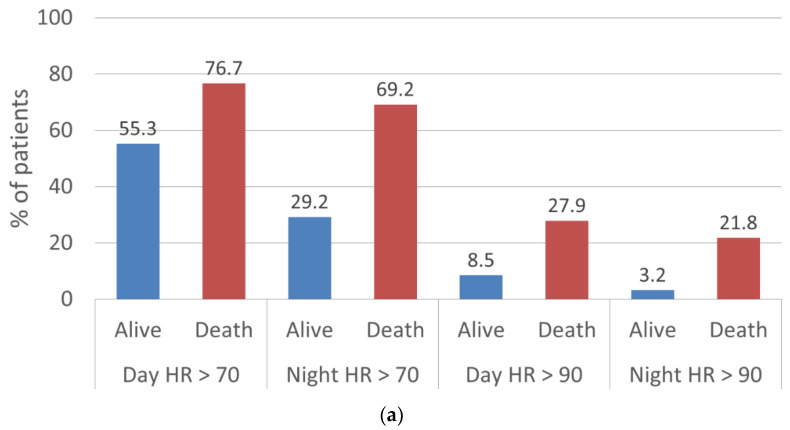
(**a**) The percentage of patient HR > 70 and HR > 90 at EOU. (**b**) Death rates as a function of nighttime HRs with a threshold of 70 bpm at BOU and EOU. (**c**) Death rates as a function of nighttime HRs with a threshold of 90 bpm at BOU and EOU.

**Table 1 jcm-11-02901-t001:** Clinical characteristics.

Total number of patients	4509
Number of patients who died	88 (2%)
Age, years	59 ± 13
Male gender	3611 (80%)
Length of use, median (IQR) (days)	66 [IQR 43–90]
Diagnosis	
Congestive heart failure	1587 (35.2%)
PCI ^1^/CABG ^2^/MI ^3^	1493 (33.1%)
Myocarditis	306 (6.8%)
ICD ^4^ explant	197 (4.4%)
Others	1054 (23.4%)

^1^ Percutaneous coronary intervention; ^2^ coronary arterial bypass grafting; ^3^ myocardial infarction; ^4^ implantable cardioverter defibrillator.

**Table 2 jcm-11-02901-t002:** Mean HR comparison of alive vs. dead.

	Daytime BOU ^1^	Daytime EOU ^2^	Nighttime BOU	Nighttime EOU
Alive, bpm (*n*)	76 ± 13 (4300)	73 ± 13 (4178)	69 ± 13 (4259)	65 ± 12 (4120)
Dead, bpm (*n*)	80 ± 15 (85)	84 ± 20 (86)	76 ± 14 (80)	80 ± 20 (78)
*p*-Value	<0.01	<0.0001	<0.0001	<0.0001

^1^ Beginning of use; ^2^ end of use.

**Table 3 jcm-11-02901-t003:** Patient outcome with different threshold levels.

	HR Threshold	Daytime BOU	Daytime EOU	Nighttime BOU	Nighttime EOU
Alive, *n* (%)	70 bpm	2823 (65.7)	2310 (55.3)	1888 (44.3)	1204 (29.2)
	90 bpm	572 (13.3)	354 (8.5)	280 (6.6)	130 (3.2)
Dead, *n* (%)	70 bpm	63 (74.1)	66 (76.7)	51 (63.8)	54 (69.2)
	90 bpm	21 (24.7)	24 (27.9)	11 (13.8)	17 (21.8)

**Table 4 jcm-11-02901-t004:** Nighttime HR trend with different thresholds.

Trend	HR Threshold	*n*	BOU, bpm	EOU, bpm	Death, *n* (%)	*p*-Value
High–high	70 bpm	899	82 ± 10	80 ± 10	34 (3.8)	<0.0001
	90 bpm	45	99 ± 9	101 ± 12	4 (8.9)	0.40
High–low	70 bpm	887	79 ± 9	61 ± 6	11 (1.2)	<0.0001
	90 bpm	220	98 ± 8	71 ± 11	5 (2.3)	<0.0001
Low–high	70 bpm	324	64 ± 5	79 ± 11	16 (4.9)	<0.0001
	90 bpm	97	74 ± 10	102 ± 14	10 (10.3)	<0.0001
Low–low	70 bpm	1968	60 ± 7	57 ± 7	13 (0.7)	<0.0001
	90 bpm	3716	67 ± 10	63 ± 10	55 (1.5)	<0.0001

## Data Availability

Data used in this study can be made available upon reasonable request.

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
