# Peer review of "High Resting Heart Rates Are Associated with Early Posthospitalization Mortality in Low Ejection Fraction Patients"

_jcm, 2022, doi:10.3390/jcm11102901_

Round 1

Reviewer 1 Report

In this retrospective study the authors after analysing a large sampe of patients evaluated the impact of HR daytime and nighttime mean values on mortality based on data retrieved from wearable WCD and found a significant association between increased HR and mortality.

Although it is a well wrtitten manuscript i find difficult to understand the novelty of their findings. The association between high HR and outcome in cardiovascular diseases is very well established. The trend of HR during a 3 months period has alos been tested n different studies. Besides WCD is not that commonly in all patients and its value for data retrieval and off line analysis is limited.

I suggest the authors should mention also circadian mortality and potential association between circadian HR fluctuations and death events, although they were extremely limited in the context of their large sample.

Regarding statistics i think they should apply more sophisticated techniques and post hoc analyses for assessing HR circadian fluctuations' impact upon mortality

Author Response

In this retrospective study the authors after analysing a large sampe of patients evaluated the impact of HR daytime and nighttime mean values on mortality based on data retrieved from wearable WCD and found a significant association between increased HR and mortality.

Although it is a well wrtitten manuscript i find difficult to understand the novelty of their findings.

Comm.: This paper describes for the first time the possibility to monitor heart rate in critically ill patients and how this could potentially influence treatment decisions.

The association between high HR and outcome in cardiovascular diseases is very well established. The trend of HR during a 3 months period has alos been tested n different studies. Besides WCD is not that commonly in all patients and its value for data retrieval and off line analysis is limited.

Comm.: The WCD delivers detailled data on the patient's heart rate over the wear period for more than 22 hours per day. While the WCD might not be commonly used in all patients, those who wear it are closely monitored and could be treated towards teh desired heart rate

I suggest the authors should mention also circadian mortality and potential association between circadian HR fluctuations and death events, although they were extremely limited in the context of their large sample. Regarding statistics i think they should apply more sophisticated techniques and post hoc analyses for assessing HR circadian fluctuations' impact upon mortality

Comm.: We do not believe the sample size is large enought to make a significant statement. Also information on patient's death is extremely limited due to the retrospective nature of this work. A proespective trial design would be ore suitable for collecting these datapoints

Reviewer 2 Report

The abbreviation "CHF" should be defined.

" To be included in the dataset, patients had to have a minimum HR recording time > 250 minutes between 7 a.m. and midnight  (daytime) or > 100 minutes between midnight and 7 a.m. (nighttime) for at least one day 
during the first three calendar days of WCD wear and for at least one day during the last  three calendar days of WCD wear." The inclusion criteria are not clear.

A photo of the device with an associated description is indicated in the "Device Description" section.

Why did the authors divide the groups into "HR > 70 bpm" and "HR > 90 bpm."

"elevated HR (> 70 bpm) ". An HR of 80 b/min is considered elevated? In my opinion, it is not. So the design of the study should be revised.

Results should be rephrased as they are hard to follow.
"High-high, High-low,  Low-high, Low-low " trends should be defined.
Table 3 and 4 could be combined into one table.
Table 5 and 6 could be also combined into one table.
Figures should also have an associated description, but I am not sure if they bring additional information.

It is still unclear how this device can help clinicians in guideline-directed medical therapy.

Author Response

The abbreviation "CHF" should be defined.

Comm.: Added to the document

" To be included in the dataset, patients had to have a minimum HR recording time > 250 minutes between 7 a.m. and midnight  (daytime) or > 100 minutes between midnight and 7 a.m. (nighttime) for at least one day 
during the first three calendar days of WCD wear and for at least one day during the last  three calendar days of WCD wear." The inclusion criteria are not clear.

Comm.: Clarified in the manuscript, the night time vs daytime discriminator is based on the reduced activity of the subject during midnight and 7 a.m. based on the measurement of the subject's activity level.

A photo of the device with an associated description is indicated in the "Device Description" section.

Comm.: added to the manuscript

Why did the authors divide the groups into "HR > 70 bpm" and "HR > 90 bpm."

Comm.: see below & added to the manuscript. >90bpm follows the conventional definition of tachycardia used in earlier studies (for example Swedberg et al, Kolloch et al…)

"elevated HR (> 70 bpm) ". An HR of 80 b/min is considered elevated? In my opinion, it is not. So the design of the study should be revised.

Comm.: As descriped 70 bpm is the target HR recommended by the ESC guidelines, rather than elevated we could use the term "above desired range".

Results should be rephrased as they are hard to follow.
"High-high, High-low,  Low-high, Low-low " trends should be defined.
Table 3 and 4 could be combined into one table.
Table 5 and 6 could be also combined into one table.
Figures should also have an associated description, but I am not sure if they bring additional information.

Comm.: Tables combined in the manuscript - caption rephrased to threshold leves - that should clarify

It is still unclear how this device can help clinicians in guideline-directed medical therapy.

Comm.: Patients wearing a WCD can be constantly monitored remotely (Zoll Patient Management Network) and GDMT (here BB) can be adapted in order to reach the desired heart rate.

Round 2

Reviewer 1 Report

The authors have addressed all reviwers' comments and suggestions

Author Response

Thank you!

Reviewer 2 Report

It is hard to follow the improvements since the authors did not highlight them.

Figures 2 and 3 do not have a proper description and neither OY axis is defined.

Author Response

We thank you for the extensive review of the manuscript and would like to answer to your comments.

“It’s hard to follow the improvements since the authors did not highlight them” We followed the guidelines of the manuscript to submit the article not in track change mode

“Figures  2 and 3 do not have proper description and neither OY axis is defined” We added the definition of the y-axis; the description can be found in the caption of each graphic.

We also want to add that the manuscript has been reviewed by a native speaker from the US.